# Integrated Sustainable Development of Culture into Tourist Map Design: A Case from Foshan, China

**DOI:** 10.3390/ijerph192114191

**Published:** 2022-10-30

**Authors:** Liting Zhou, Fei Ouyang, Yan Sun, Wentao Chen, Yiyong Li, Ruyu Zhao

**Affiliations:** 1School of Urban Culture, South China Normal University, Guangzhou 510631, China; 2School of Fine Arts, South China Normal University, Guangzhou 510631, China; 3School of Geography, South China Normal University, Guangzhou 510631, China

**Keywords:** cultural sustainability, tourist map, information design, historical city, human wellbeing

## Abstract

As the fourth pillar of sustainable development, culture is widely recognized as contributing to human wellbeing. The distinctive culture of cities is an important driving force for attracting visitors to destinations for tourism consumption. Since historical cities have important cultural and historical values, the design of their tourist maps needs not only geographic positioning and artistic aesthetics, but also a systematic design method to present the connotation of regional cultures, so as to enhance the local cultural identity of hosts and the cultural cognition of visitors, and to drive the local tourism economy, improve the regional environment, promote cultural transmission and inheritance with the help of tourist map design in terms of cultural sustainability, which ultimately achieves sustainable development of human wellbeing. Taking Foshan, a national historical city, as an example, combined with the cultural gene and the cultural hierarchy theory, this study analyzes and summarizes the regional culture of Foshan from three aspects: material cultural gene, intangible cultural gene and spiritual cultural gene. This study also comprehensively presents the geographical information and historical or humanistic characteristics of the city through direct translation, narrative translation, and metaphor translation, which provide theoretical support and practical guidance for the integration of regional cultures into tourist map design.

## 1. Introduction

Cultural sustainability has become a growing important priority in the sustainable development agenda, and is now often described as the fourth pillar, equal to social, economic, and environmental issues which affect human wellbeing [1]. It reflects a shift in human development goals from a focus on material wellbeing, such as income, to a focus on humanities development and quality of life [2]. As a significant force on the reproduction of culture and the transmission of cultural values, design is an important means for cultural innovation, creation, and entrepreneurship. Nowadays, design thinking has shifted from an emphasis on functional design to a stress on emotion, culture, and memory [3]. Conceiving the design of tourist maps, based on this new concept, offers new ideas for cultural sustainability. As a key tool to provide urban tourism information, tourist maps can play a positive role in inheriting and disseminating regional cultures and promoting the sustainable development of culture [4]. In addition, the integration of regional cultural elements into the design of tourist maps is not only conducive to cultural transmission with “visitors” as a carrier, but also helps to enhance the host’s sense of identity with their own culture, and finally promotes human wellbeing through the tourist’s satisfaction and the host’s happiness.

Affected by the COVID-19 pandemic, people’s physical and mental health were harmed to varying degrees, while tourism has a positive effect on their recovery [5,6]. People are usually riveted by specific cultures, traditional communities, and healthy service activities of tourist attractions, and they are integrated into the local humanistic environment through experiences that free the mind and body [7]. With the steady increase in national income, the public’s demand for the improvement of wellbeing has become increasingly urgent. For the tourism industry, which ranks first in the “Five Happiness Industries”, the integration of sustainable development of cultural into tourism has become a fundamental goal to improve the public’s wellbeing [8]. At present, research on the integration of cultural sustainability and tourism to promote human wellbeing mainly adopts quantitative methods, such as the coupling coordination model, to measure the integration of cultural industry and tourism industry in a narrow sense [9]. In a broad sense, it discusses the integration of more macroscopic cultural atmospheres and cultural elements with tourism products, such as the integration of ethnic festivals and tourism performances, the integration of social cultural memory and destination tourism, and the integration of traditional culture and modern ecotourism [10,11]. And, from a micro perspective, it discusses the improvement of tourist satisfaction, loyalty, and other tourism experiences [12]. However, previous studies have mainly adopted quantitative or qualitative methods, and there are few mixed studies that integrate cultural sustainability, tourism, and map design. As the preservation of cultural heritage and the promotion of cultural vitality have been identified as keys to achieve cultural sustainability [13], the integration of regional cultural elements into tourist map design appears to be an important way to promote the sustainable development of regional cultures. Therefore, this study attempts to integrate the sustainable development of culture into tourist map design from an interdisciplinary perspective. So far, only a few studies have explored the translation and representation of individual aspects of a regional culture and its informational elements on tourist maps to date [14,15], and little attention has been paid to design strategies for presenting regional cultural elements, scientifically and systematically, in tourist maps [16]. Some scholars have pointed out that promoting cultural cognition of tourist destinations ensures higher subjective satisfaction for tourists [17,18,19]. Moreover, the scientific and systematic presentation of a regional culture on the tourist map also helps to enhance the host’s cultural identity and aesthetic, and the economic benefits brought to the local area by visitors’ cultural tourism can increase the driving force for the inheritance and protection of excellent local culture, which in turn promotes the local government and residents to improve the regional environment through modeling of the cultural tourism city. Thus, the integration of regional cultural elements into the design of urban tourist maps is of great importance to improving the wellbeing of residents and tourists.

Since the reform and opening up in China, historical cities have gradually become important destinations for domestic and foreign tourists because of their rich cultural heritage and their function of preserving the customs, memories, and identity of communities [20]. However, due to the lack of full use of their own cultural characteristics, some historical cities have caused the problem of “homogenization” on tourist maps in the process of urban image transmission, which may have a negative impact on tourists’ understanding of local culture and the planning of tourism routes through the maps [16,21]. Therefore, we can improve the design of tourist maps to better present the native culture and to promote visitors’ culture cognition.

At present, the design of a tourist map mainly includes two types: the traditional map and the hand-drawn map. Traditional tourist maps pay attention to the scientific nature of map design, but the function of this kind of map often stays in meeting the demand of positioning for visitors. Furthermore, too many road details will increase the cognitive burden for them. For example, traditional maps often use general geometric symbols to represent the location of various scenic spots, which reduces the ability and efficiency of visitor recognition [22]. Hand-drawn maps pay more attention to the artistry of map design, and usually carry out special symbol design for each scenic spot in the form of hand-drawn visuals, so as to make the tourist map meet the aesthetic demand for visitors, and thus play an important role in displaying local culture and improving urban image [22,23]. However, within the scope of the relationship between tourism and culture, cultural resources are the foundation and have become an important force in attracting domestic and foreign tourists to their destinations [24]. Therefore, some scholars have proposed that the design of tourist maps should not only take into account the scientific and artistic aspects, but also fully consider the presentation of culture in the tourist map [16]. However, a historical city’s regional culture can be divided into multiple categories. Most tourist maps focus on reflecting the material culture of the city, especially the scenic spots and local delicacies directly related to tourism. There is a lack of a systematic expression framework and translation methods for a regional culture, and the narrative dimension is single, which is not conducive to the comprehensive presentation and expression of the regional cultural characteristics of historical cities. Hence, taking Foshan City, a national historical city in China, as an example, this study aims to: firstly, construct the expression framework (using map design) of a regional culture of the city based on the cultural gene and the cultural hierarchy theory; secondly, excavate and establish the local cultural pedigree of the city; and finally, use the translation method to realize the combination of different classified cultural elements and the design of a tourist map so that it is expected to provide a reference for the exhibition of regional cultural characteristics of historical cities in the map.

## 2. Materials and Methods

### 2.1. Research Area

Foshan City, called Jihua Township in ancient times, originated during the Jin Dynasty and was named after the Tang Dynasty. It has a history of about 4500 to 5500 years, and it is a national historical city in China [25]. Foshan City is located in the hinterland of the Pearl River Delta of the Guangdong Province in southern China, adjacent to the capital city of Guangzhou to the east, and belongs to the distribution area of Cantonese culture, an important branch of Lingnan culture. Historically, Foshan was famous for its ceramics, textiles, casting, and medicine in China. Due to its developed handicraft industry and prosperous commerce, Foshan was known as one of the four famous towns, along with Jingdezhen, Hankou, and Zhuxian, in the Ming and Qing Dynasties. Together with Beijing, Wuhan, and Suzhou city, it is also known as one of the four major commercial gathering places. What is more, it is known as the hometown of pottery art, martial arts, and Cantonese opera, so that its rich cultural heritage has laid the foundation for Foshan as an important birthplace of Lingnan culture. In 2020, the population of permanent residents in Foshan was 9.52 million, and there are 17 national 4A- or 5A-level tourist attractions, which are generally considered to have good service quality and are worth visiting, and 14 national intangible cultural heritage attractions [26,27], among which Cantonese Opera has been categorized as an Intangible Cultural Heritage of Humanity by UNESCO [28]. Foshan’s profound historical and cultural heritage attracts domestic and foreign tourists. In 2019, it received 2.16 million international tourists and 60.10 million domestic ones, with a total tourism revenue of 89.186 billion yuan [29]. The geographical location of Foshan can be seen in Figure 1.

### 2.2. Cultural Pedigree and Translation Method

Before presenting the cultural elements of the historic city on a tourist map, we need to systematically classify the regional culture and establish its pedigree in order to adopt a suitable method to translate it onto the tourist map [30]. This study used the cultural gene theory originated by Richard Dawkins, a biologist from Oxford University. Inspired by biological genes, he suggested that culture is composed of memes and believed that the meme is the basic unit of cultural transmission or inheritance in different periods [31]. Chinese academia often paraphrases “meme” as “cultural gene”, and thus embarks on a relatively independent research and development path [32]. Influenced by Dawkins, Chinese scholars Liu Changlin [33], Wu Qiulin [34], Zhao Chuanhai [35], Bai Guixi [36], and others have defined the concept of the cultural gene from different perspectives. Overall, cultural genes are factors that play a decisive role in the nature and characteristics of a cultural system and have reproducibility and relative stability [37]. The cultural gene of a city is a measure of the city’s locality. The unique regional culture of a city is the evolution process from a non-cultural space to a cultural place after a long period of human activities and the interactive influence of specific regions, and gradually presents the culture as different from other regions [38]. As the basic unit of the regional cultural inheritance system, cultural genes make different cities emerge different cultural features in the process of historical development [31,39]. Scholars have constructed a variety of classification methods for cultural pedigrees according to different objects of concern, including historical and natural culture genes, visible and invisible culture genes, material and intangible culture genes, as well as subject genes, attachment genes, mixed genes, and variant genes [21,40]. According to the tourist map design of historical cities in this study, the local cultural heritage in material and intangible forms is an important content of tourists’ attention, and the cultural heritage and the cultural vitality are also identified as keys to achieve cultural sustainability [13]. Therefore, this study divides the local cultural pedigree of historical cities into material cultural genes and intangible cultural genes. Material cultural genes are mostly exhibited and spread in the form of materials, while intangible cultural genes usually refer to non-material forms of culture inherited through oral narration or behavioral expression with artistic or historical value [41].

He proposed that any culture can be divided into three levels: surface, middle, and deep [42]. Consistent with this view, Malinowski, a cultural anthropologist, divides culture into the material level, the institutional level, and the spiritual level [43]. On the basis of the three structures of culture, some scholars further propose three levels for the design and application of traditional cultural products: the material–symbol level, the behavior level, and the internal spiritual level [44]. According to the above levels, it can be concluded that the material culture gene belongs to the material level, the intangible cultural gene pertains to the behavior level, and the spiritual level, which reflects people’s value orientation and represents the deepest core of culture, is the cornerstone of regional culture. In view of this, the present study combines the cultural hierarchy theory to further expand the classification of cultural genes and divides it into three dimensions: material cultural genes, intangible cultural genes, and spiritual cultural genes. Based on the above classification, the cultural pedigree of the historical city is as Figure 2.

Despite the numerous followers of Richard Dawkins’ cultural gene theory, genetic models of analogous biology are insufficient to provide a functional framework for research [45]. Therefore, in the disciplinary context of map information design, it is necessary to construct interdisciplinary expressions for the presentation of regional culture. The term “translation” is a linguistic term that originally refers to the act of using another language to translate the original text of a certain language [14]. In the process of practice, “translation” has gradually broken through the realm of linguistics and has been widely used in the fields of craft design, architectural design, and cultural and creative product design [46,47], so it has an important methodological value. Its theoretical concept has been further extended as “the process in which one set of ideographic systems influences the generation of another set of ideographic systems with certain rules” [48]. Therefore, this study attempts to combine the cultural gene theory for cultural sustainability, the design of tourist maps in the field of cartography, and the translation method in the domain of linguistics, and on the premise of ensuring the accuracy of the tourist map, to explore the exhibition method of integrating the regional culture of historical cities into the tourist map. The translation methods corresponding to the cultural pedigree in the tourist map are shown in Figure 2.

### 2.3. Data Preprocessing

#### 2.3.1. Establishing the Cultural Pedigree of Foshan

As shown in Figure 2, based on the basic functions of tourist map, this study establishes the cultural pedigree of Foshan according to three categories: material cultural genes, intangible cultural genes, and spiritual cultural genes.
Foshan’s material cultural genes include: the Ancestral temple, the Nanfeng Ancient Stove, the Qinghui Garden, and other historical sites and local specialties, such as Shiwan fish rot, Xiqiao flatbread, and Jiandui.UNESCO considers intangible cultural heritage as that which “includes traditions or living expressions inherited from our ancestors and passed on to our descendants, such as oral traditions, performing arts, social practices, rituals, festive events, knowledge and practices concerning nature and the universe or the knowledge and skills to produce traditional crafts”. Therefore, this study suggests that Foshan’s intangible cultural genes include: culture and entertainment activities (Cantonese Opera, the Woodblock New Year Picture, Ten-fan Music, and the Autumn Scenery Festival, etc.), and handicrafts and skills (traditional handicrafts and Kung Fu martial arts).In terms of spiritual culture, Foshan was called Jihua in ancient times. Because three Buddha statues were excavated in Tapogang in the city during the Tang Dynasty, the people thought it was the territory of Buddhists, so Jihua was changed to “Foshan”, with “Chan” as the abbreviation. The historical origin of the city and Buddhist culture can be seen. In addition, Foshan residents retain the custom of “traveling Tongji” during the Lantern Festival every year. The Tongji Bridge next to the Ancestral Temple means “you need to cross the bridge and then you will benefit”, and the folk activity of “Crossing Tongji” pines people’s yearning for a better and harmonious life. Furthermore, Foshan was one of the four famous towns in China during the Ming and Qing Dynasties, and it was also the birthplace and prosperous space of Lingnan culture. However, the terrain was flat, and the natural defense conditions were relatively poor. In order to protect themselves, the residents formed the tradition of practicing martial arts and strengthening their health early. What’s more, Huang Feihong, Yip Man, and Bruce Lee further developed Foshan Kung Fu. Therefore, the spiritual cultural gene of Foshan also includes cultural characters and spiritual traits, such as Tongji harmony and advocating Confucianism and martial arts.

#### 2.3.2. Determination of Map Extent of TPOI

In order to determine the mapping area of the tourist map, this study collected the locations of tourist attractions in Foshan. Figure 3 is generated by buffer analysis based on the distribution of the attractions. Among them, more than 60% of the attractions are concentrated within 20 km of the Ancestral Temple. To improve the ability and efficiency of tourists to identify scenic spots, this study placed the Ancestral Temple in the center of the main map on the front of the tourist map and chose a scale of 1:200,000 to better present most of the tourist points of interest (TPOI).

## 3. Results

### 3.1. Direct Translation of Material Cultural Genes

As shown in Figure 4, in the information design of the tourist map, the direct translation method can be used for the material cultural genes with regional characteristics. For example, when translating typical buildings and local specialties in scenic spots, it is only necessary to imitate and copy the original shape and color of the object and summarize and exhibit its visual characteristics through the information design language to form a recognizable image symbol. In addition to tourist attractions and regional delicacies, physical infrastructure such as accommodation, transportation, and shopping is also important information that tourists want to obtain. However, as some of the main information on tourist maps, the information design of the urban material layer is different from the administrative division map in that it is more pertinently presented around the information of scenic spots and related tourism infrastructure. Therefore, it is necessary to form a favorable visual perception of material cultural genes by adjusting the color saturation, size of the symbols, and the thickness of the line. The material symbol and visual design can be seen in Figure 5.

### 3.2. Narrative Translation of Intangible Cultural Genes

#### 3.2.1. Combined with the Exhibition of History and Space

As shown in the section “Special Intangible Heritage” in the back of the map, in the history dimension, through the symbolic exhibition of traditional cultural and entertainment activities, handicrafts and the story expression of the main content of traditional martial arts, the reproduction of typical intangible cultural genes and their representatives will help the tourist to understand the intangible cultural content such as traditional featured festivals, crafts, and music of the historical city. Through the introduction of background texts, the narrative translation of intangible cultural genes is realized by means of “words plus (+)” translation of intangible cultural scenes and representative characters.

Many intangible cultural heritages take the material entity as the carrier and survive with it. Some museums, exhibition halls, memorial halls, theaters, and other places can often display intangible forms of culture for tourists. Therefore, in the spatial dimension, it is necessary to determine the typical material carrier and geographical location of this part of intangible cultural genes and present the geographical space of intangible cultural genes on the tourist map. The example of spatial dimension is shown in the section “Itinerary of Martial Art” in the back of the map.

#### 3.2.2. The Exhibition of Theme Itineraries

Historical cities usually have abundant intangible cultural heritage, but in the process of tourist map design, intangible cultural heritage often lacks themed itineraries. The intangible cultural tourism projects of historical cities are connected in the form of theme tourism routes, which can easily bring tourists a better intangible cultural tourism experience. Foshan is a famous city for Kung Fu, and it is the home of Cantonese opera. Therefore, this study intends to systematically exhibit the intangible cultural genes of different themes by designing the “Itinerary of Martial Art” and the “Folk entertainment themed itinerary”. For the key areas of intangible cultural tourism projects, the method of changing the scale is used to optimize the visual exhibition of tourism projects. As shown in the section “Itinerary of Martial Art” of the back of the map, a large-scale map of each intangible cultural site is shown next to the image symbols of the site, and a small-scale map containing the theme itinerary is shown on the far right of the map.

### 3.3. Metaphorical Translation of Spiritual Cultural Genes

The metaphor was originally a rhetorical method in linguistics. With the continuous development of research fields such as cognitive and behavioral psychology, metaphor is gradually considered to play an important role in people’s cognitive process, reflecting people’s understanding of the world [49]. George Lakoff and Mark Johnson point out that the essence of metaphor is the cognitive phenomenon, that is, understanding and experiencing an object through other objects [50]. As a design method, metaphor can fully explore the meaning of a culture and convey the inner spirit of it. Moreover, Qin and Ng proposed to use metaphors to explain the cultural meaning, not only by taking into account the sustainable development of culture, but also by preserving local cultural characteristics [51].

Buddhist culture pays attention to the integration of humans and nature, and the colors representing nature, life, and holiness can better reflect the charm of Buddhism. Through the refining of Foshan’s spiritual cultural genes, it is recognized that Foshan’s regional culture also contains the spiritual characteristics of pursuing harmony and advocating Confucianism and martial arts. At the same time, the spiritual characteristics of the city will also be affected by the regional environment. Foshan has a typical subtropical Lingnan water town classification and forest landscape. Therefore, Foshan’s unique cultural character and natural environment give people a simple, calm, soft, and full-of-vitality impression, and this invisible impression belongs to the metaphorical information part, which can be expressed through color and graphic symbols in the design of tourist maps.

#### 3.3.1. Color Metaphor

In terms of color information transmission, color, as a unique visual element, can convey different emotions and give people different color impressions [14]. Through the above spiritual cultural analysis, when choosing the main design tone of Foshan tourist map, it will tend to use dark green, representing freshness, elegance, and calmness. In terms of secondary color, it is also emphasized to give priority to cool colors to unify the rational and high-end feelings. Therefore, blue, an adjacent color of green, is used as the color to frame the extent and distinguish between different modules of the map to maintain the consistency of emotional characteristics. Conversely, red is used as a direct contrasting color of green for emphasis in key tourist attractions. This kind of red is also the symbol color of Foshan’s ancient buildings. Moreover, the map uses neutral colors as spacing methods, such as light gray, black, and white, to adjust the overall color density. Finally, this study adjusts the freshness and saturation of the color according to the layer level and cooperates with the color of the font to form a good visual effect of color language. The system of metaphorical color is shown in Figure 6.

#### 3.3.2. Graphic Symbol Metaphor

In terms of graphic symbol design, Foshan has a certain historical relationship with Buddhist culture. In addition, Foshan’s typical Lingnan water town scenery also makes people feel that they are in a Chinese ink painting. Therefore, in the design of the title symbols of the tourist map and the scene symbols of the intangible culture, the “blank space” and “freehand brushwork” techniques of ink painting are used for reference, and the details of the scene and its characters are not paid attention to, which give the visitor room for understanding and imagination in the blank space, and reflect profound Buddhist thoughts. The graphic symbol metaphor is shown in Figure 7.

### 3.4. The Tourist Map of Foshan

As shown in Figure 8, the size of the tourist map of Foshan is 420 × 297 mm. The front map is mainly divided into four parts, showing the map space of material cultural genes, the text space of material cultural genes introduction, the tourism traffic map, and the topographic map of historical cities. The map on the back uses the F-vision rule of scanning the page from left to right and top to bottom [52], and the puzzle narrative method, to present the thematic paths and symbols of the intangible cultural tourism projects, and adds text space next to the symbols of intangible cultural genes and their representatives to help tourists understand the regional culture of Foshan. To complete the design of the tourist map, ArcMap 10.7 was used for spatial analysis and map production; Photoshop 2020, Adobe Illustrator 2019, Procreate, and Affinity Designer were used for drawing hand-drawn symbols and map designs. Ultimately, the final tourist map design results are shown in Figure 9 and Figure 10.

## 4. Discussion

Although the integration of cultural sustainability and tourism have attracted extensive attention from scholars [9,11,12], this study further broadens its research perspective and scope by introducing the tourism map design in the field of cartography and the translation method of linguistics into the presentation and expression process of regional culture. It proposes a framework for designing urban tourist maps based on cultural gene theory and cultural translation methods for designers or researchers to enhance the native cultural identity of hosts and the cultural cognition of visitors, which aims to drive the local tourism economy, improve the regional environment, promote the transmission and inheritance of regional culture, and ultimately achieve the sustainable development of human wellbeing through integrating regional culture into tourist map design. Despite a few previous studies about integrating culture into map design [14,22,53], they tend to focus on the presentation and expression of material culture, while immaterial and spiritual cultural elements have rarely been applied to the design of tourist maps [15,16,44]. Therefore, we drew on the cultural gene theory and further broadened the classification of urban cultural genes based on the cultural hierarchy theory from the three levels, to establish the cultural pedigree of cities in terms of material cultural genes, intangible cultural heritage genes, and spiritual cultural genes, which can help promote the systematic integration of native cultural elements into the design of tourist maps.

Language is a part of culture and reflects it. In fact, language plays an important mediating role in the transmission, inheritance, and sustainable development of culture [54]. Therefore, this study applies the linguistic method of “translation” to the design strategy of integrating regional culture into tourist maps, and adopts direct translation, narrative translation, and metaphor translation methods corresponding to the cultural pedigree of historical cities, which provide a functional methodological framework for the presentation of regional culture on tourist maps [45]. Among them, for the presentation of material cultural elements, the direct translation method has been applied to the design of most hand-drawn tourist maps [14,16], while the presentation of intangible cultural elements has received less attention [15]. This study explores the application of the narrative translation method to provide design ideas for tourist maps by combining the same with the exhibition of history, space, and theme itinerary. In addition, the metaphor design method is rarely used as a way to facilitate visitors’ perceptions of cultural connotations [51], and only a few studies have applied it to the design of traditional cultural products, modern cultural and creative products, or architectural landscapes [51,55]. Therefore, this study explores design in terms of the color metaphor and the graphic symbol metaphor with the aim of providing methodological references for the presentation of spiritual cultural elements in tourist maps. What’s more, in recent years, many cities have introduced programmatic policy documents to promote the sustainable development of culture [56]. The design model and methodology proposed in this study provide theoretical support and practical guidance for integrating sustainable development of culture into tourist map design.

## 5. Conclusions

According to the significance of the exhibition of regional culture in tourist maps, this study proposes a regional cultural expression framework and exhibition method for tourist maps based on cultural gene theory and cultural translation methods. Taking Foshan, a famous national historical city in China, as an example, the typical regional culture genes of Foshan are collected, and its pedigree is constructed. Then, the scale and mapping area of the Foshan tourist map is determined by buffer analysis. Aiming at the material cultural genes of Foshan, this study adopts the method of direct translation to summarize and imitate the original shape and color of the objects. In terms of intangible cultural genes, this study adopts the narrative translation, including tourism theme line design and integration of history and space exhibition. In addition, this study adopts the translation method of color metaphor and graphic symbol metaphor for spiritual cultural genes. These methods can also be applied to the regional cultural translation and exhibition of tourist maps of other historical cities so as to integrate sustainable development of regional culture into tourist map design. The limitation of this study is that it ignores the subjective feelings of the audience and does not use quantitative analysis methods to evaluate the substantial effect of the expression framework on the audience’s understanding of the historical city tourist map and its regional culture. Therefore, the next step of the study can consider the application of individual interviews, the analytic hierarchy process, or eye movement experiments to carry out follow-up tourist map adjustment or design work.

## Figures and Tables

**Figure 1 ijerph-19-14191-f001:**
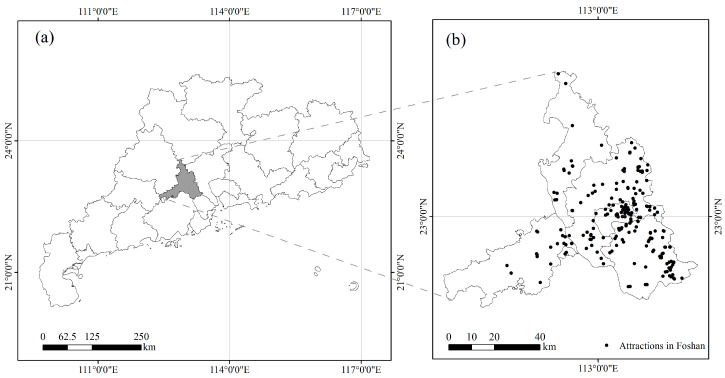
Geographical location of Foshan: (**a**) the location of Foshan City in Guangdong Province; (**b**) distribution map of major scenic spots in Foshan.

**Figure 2 ijerph-19-14191-f002:**
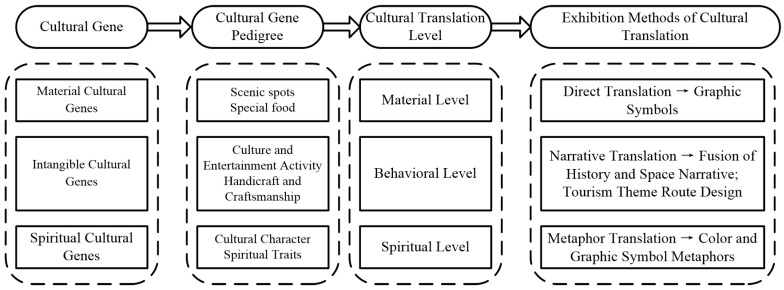
Cultural pedigree and translation methods for regional culture.

**Figure 3 ijerph-19-14191-f003:**
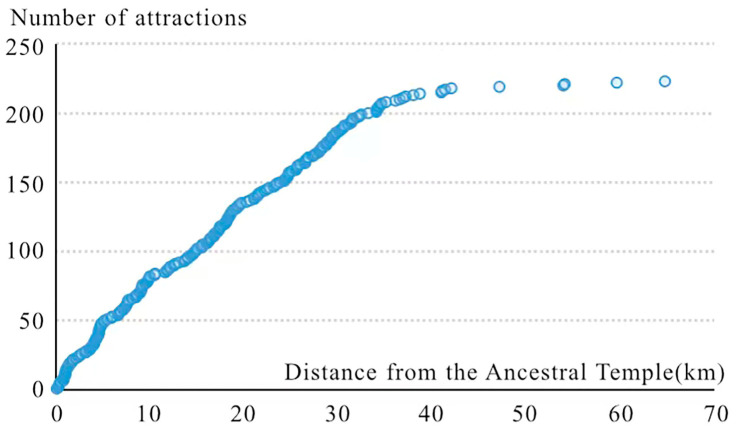
Number of attractions with distance.

**Figure 4 ijerph-19-14191-f004:**
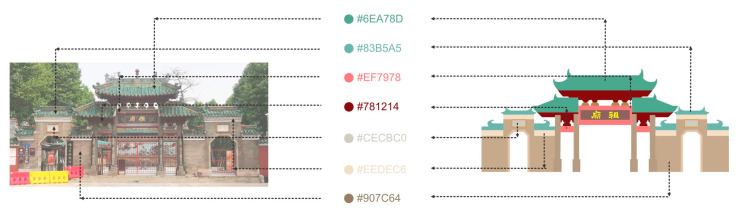
Color extraction of tourist attractions (Ancestral Temple).

**Figure 5 ijerph-19-14191-f005:**
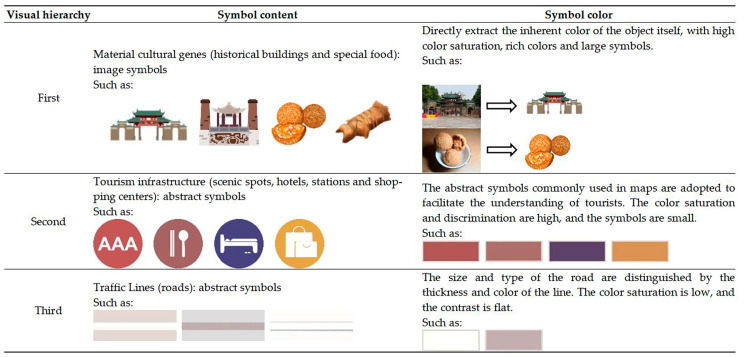
Material symbol and visual design.

**Figure 6 ijerph-19-14191-f006:**
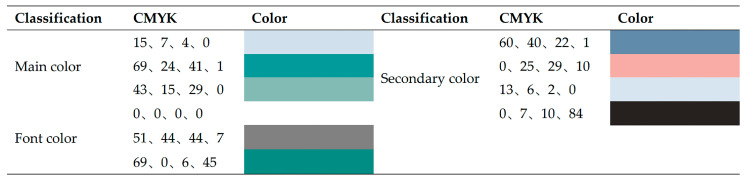
The system of metaphorical color.

**Figure 7 ijerph-19-14191-f007:**
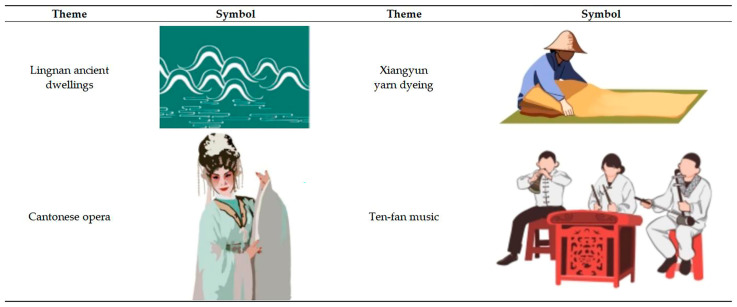
Graphic symbol metaphor.

**Figure 8 ijerph-19-14191-f008:**
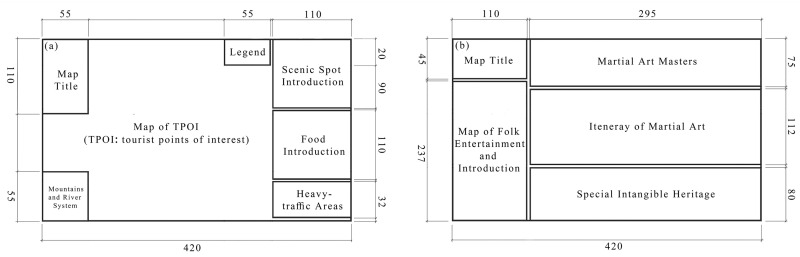
The layout design and expression objects of the tourist map of Foshan: (**a**) front of the map; (**b**) back of the map.

**Figure 9 ijerph-19-14191-f009:**
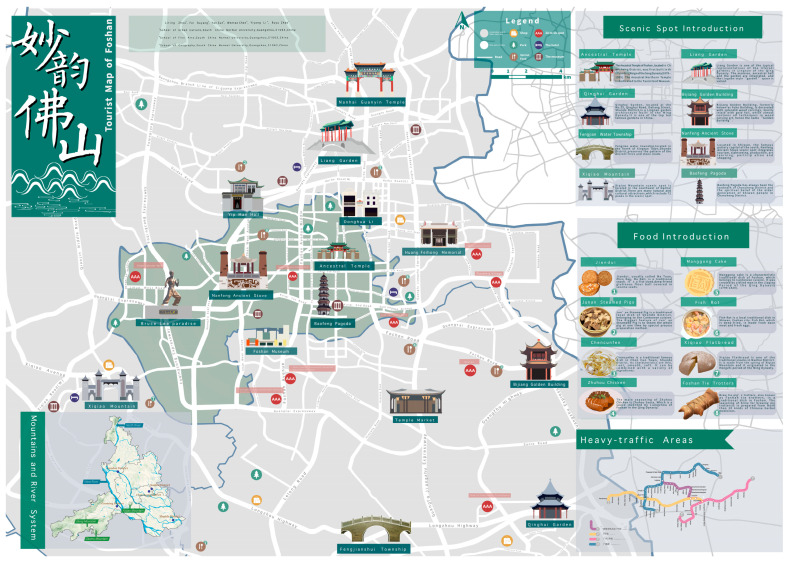
Tourist map of Foshan (Front of the map).

**Figure 10 ijerph-19-14191-f010:**
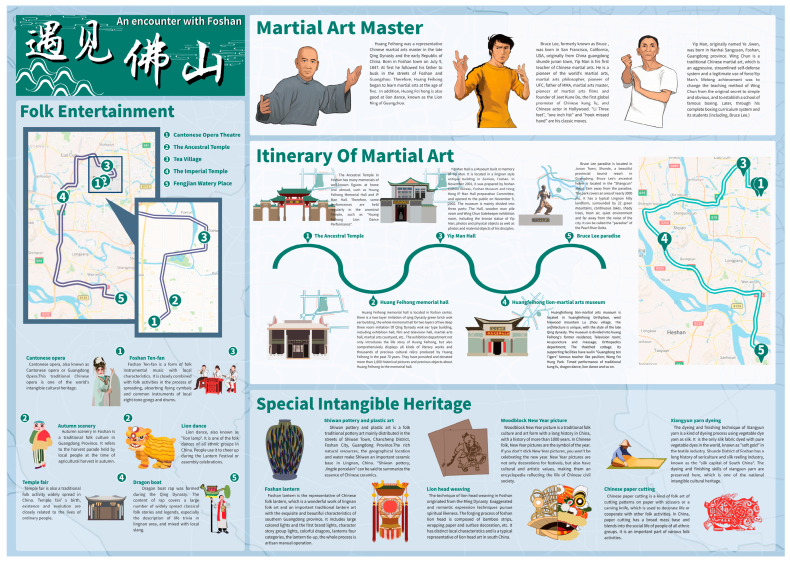
Tourist map of Foshan (Back of the map).

## Data Availability

All data were derived from the following resources available in the public domain: Aliyun Map (http://datav.aliyun.com/portal/school/atlas/area_selector, (accessed on 15 March 2022)), BaiDu Map (https://map.baidu.com/mobile/webapp/index/, (accessed on 16 March 2022)), Open Street Map (https://www.Openhistorical map.org/, (accessed on 16 March 2022)), and Mapbox (https://www.mapbox.com/, (accessed on 18 March 2022)). The data used are freely available for download from this source.

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
