# Peer review of "Integrated Sustainable Development of Culture into Tourist Map Design: A Case from Foshan, China"

_ijerph, 2022, doi:10.3390/ijerph192114191_

Round 1

Reviewer 1 Report

Dear Authors

Please see the attached file for comments

Author Response

Q1. If you are presenting that this indeed is the case, then there needs to be much greater clear explanation of the terms that you have arrived at – the material cultural gene and the intangible cultural heritage gene. Arguably it is not an historic city that is made up of cultural genes, it is made up of cultural practices, cultural influences and cultural skills and knowledge, but this is different from cultural genes. The same applies to the term spiritual gene – it is, arguably, the knowledge and learning about spirituality that is important not the inheritance of a spiritual gene. The words intangible heritage, material, tangible heritage and spirituality are of course indeed very important in the development of cultural tourism but linking them to genes is hard to make sense to the reader.

R1. Based on your suggestion, we have further elaborated on the cultural gene theory in 2.2. The Cultural Pedigree and Translation Method section. Many scholars have carried out further exploration based on Dawkins' meme theory, and summarized the concept and classification model of cultural genes.

Q2. Check whether the journal can indeed be published in colour print.

Suggest that some of the tables are presented as Figures rather than Tables – as they are more illustrative than written as a table with grid lines.

R2. We checked the papers published by IJERPH and confirmed that the journal can indeed print papers in color. In addition, we replace some tables with figures for the convenience of readers.

Q3. Table 1: State the meanings of the symbols for clarity as some are not clear such as AAA? And the luggage?

R3. The symbol with “AAA” represents tourist attractions with good quality because the letter "A" is usually used in China to mark the quality level of tourist attractions. And the “luggage” refers to the Shopping spots. Moreover, we have introduced the meaning of each symbol in detail in the Legend section of the tourist map (front of the map). We have submitted the high-definition figures to the editorial department as supplementary documents so readers can view the maps clearly.

Q4. In places throughout the article there is a confusion between the terms intangible and tangible cultural heritage as defined by the UNESCO reference used. The article would benefit from explaining that Material Cultural Genes can translate as Tangible Heritage, which would also help to link with the discussions of Intangible Heritage.

R4. Thank you for your suggestion. According to UNESCO's definition of tangible heritage, it mainly includes monuments, complexes and sites. However, according to the cultural hierarchy theory, the surface culture usually includes the form of people's use of materials, which is reflected in the field of "food, lodging, transportation, entertainment, tourism and shopping" of tourists, so we think that the use of material cultural genes is more accurate.

Q5. Table (Figure?) 2: represents tangible heritage not intangible heritage, all the illustrations are of products that are tangible, although the craft skills required to produce the artefacts are indeed intangible heritage. Some of the pictures used in the article demonstrate the act of producing a tangible object of heritage – i.e weaving – produces cloth. The cloth itself is tangible heritage. This should be made more clear throughout the article. The itinerary of Martial Art also represents tangible heritage buildings.

R5. Thank you for your suggestion and we try to make them more clear in the section 3.2. Narrative Translation of Intangible Cultural Genes.

Q6. Table 3: Identify the meaning of the colour used – but do check that the journal can print in colour as many people print or read articles in black and white print.

R6. We add the further explanation in the section 3.3.1. Color Metaphor. In addition, we checked the papers published by IJERPH and confirmed that the journal can print papers in color, so the color we use can be clearly identified in the electronic PDF. In addition, we also submitted the high-definition figures to the editorial department as supplementary documents for the convenience of readers.

Q7. Check correction of English Grammar phrases throughout such as below examples: Section 1 last paragraph – a tourist map mainly includes….: the traditional map and hand-drawn map....this study aims to: Firstly, construct the expression framework (using map design) of a regional culture of the city and …

Foshan City……, originated in the Jin Dynasty…..population of permanent residents in Foshan…p.5 - ..placed the Ancestral temple in the centre of the main map…..

R7. Thank you for your detailed suggestions. We check correction of English Grammar phrases throughout and correct them in the corresponding sentences.

Q8. Explain acronym TPOI on page 9.

R8. We have explained the meaning of TPOI (tourist points of interest) in the section 2.3.2. Determination of Map Extent of TPOI on page 6. To promote readers' understanding, we add the explanation in Figure 8.

Q9. p.5 The term “sorting out” which is used throughout the paper – should perhaps be replaced with the word “analyse” as to sort out is quite colloquial and not very academic. i.e. 2.3 .1 Analysing the Cultural Pedigree of Foshun – Or in some cases the word establishes” would be useful – this study establishes the cultural pedigree of Foshan according to three categories.

R9. We totally agree with your suggestion and correct the term in the corresponding sentences.

Q10. p.10 – grammar review – it is obtained? (it is recognised?) that Foshun’s regional culture….

R10. According to your suggestion, we replaced the obtained with recognized in p.10. And we checked the English language & style again.

Reviewer 2 Report

The designing of tourist maps that highlights the cultural essences of a region is significant to present, promote, and transmit the culture of the place. Drawing inspiration from the cultural gene and cultural hierarchy theory, this study introduces a framework for designing tourist maps of historical cities. Taking Foshan city of China as an example,  it presents a method of identifying the material, intangible, and spiritual cultural genes of the city and integrating the cultural elements of the city into the design of tourist maps. In these regards, the study is significant in providing theoretical and practical implications for exhibiting regional culture in tourist map design. The introduction of the associated background, the conceptualization of the theoretical framework, the utilization of research methods, and the presentation of research findings are all solid. From the perspective of academic quality, the paper can be accepted in its current form. The academic quality of this paper is high and solid. From this perspective, it may be accepted in its current. However, it looks that the topic of this paper is a bit deviating from the scope of IJERPH.

Author Response

Q1. However, it looks that the topic of this paper is a bit deviating from the scope of IJERPH.

Thank you very much for your approval and comment. We submitted our manuscript to IJERPH's Special Issue "Sustainable Development of Key Areas for Human Wellbeing". This Issue welcomes papers focusing on topics including (but not limited to) the following: the space of everyday life (urban and rural); the space of leisure and recreation (assets and challenges in the development of space for tourism, including space in health resorts, sustainable tourism, agritourism); spaces of high natural and cultural value (conservation threats and challenges). Moreover, culture, as the fourth pillar of sustainable development, is widely recognized as contributing to human wellbeing. Therefore, We think that the research is very consistent with the scope of this journal.

According to your suggestion, we've checked the English language & style again and minor revised for the paper.

Reviewer 3 Report

This paper proposes a framework for designing urban tourist maps based on cultural gene theory and cultural translation methods. Overall, it's valuable and interesting. I think it may be accepted, but need some improvement. There are some problems to be improved.

1 The literature needs to be improved and enhanced, the logic of the literature is not compact enough, and the cited literature needs to be transformed based on this paper, otherwise the review of the main content will appear to skip a bit.

2 Why did the author combinate of the theory of cultural gene and cultural hierarchy theory is not sufficiently demonstrated. There doesn't seem to be anything unique about this cultural classification scheme. So The author needs to clarify why this cultural pedigree system was adopted, and what’s the new.

3 The theoretical contribution of this manuscript needs to be further explained.

Author Response

Q1. The literature needs to be improved and enhanced, the logic of the literature is not compact enough, and the cited literature needs to be transformed based on this paper, otherwise the review of the main content will appear to skip a bit.

R1. According to your suggestion, we tried to improve our literature in section 1. Introduction and 2.2. Cultural Pedigree and Translation Method.

Q2. Why did the author combinate of the theory of cultural gene and cultural hierarchy theory is not sufficiently demonstrated. There doesn't seem to be anything unique about this cultural classification scheme. So the author needs to clarify why this cultural pedigree system was adopted, and what’s the new.

R2. Thank you for your suggestion. We revised our manuscript in paragraph 2 of section 2.2. Cultural Pedigree and Translation Method.

Q3. The theoretical contribution of this manuscript needs to be further explained.

R3. Based on your second suggestion, we tried to elaborate on the  Introduction and Discussion section in further details.

Reviewer 4 Report

Having as a starting point the cultural gene theory and the cultural hierarchy, the paper uses the translation method to realize the combination of different classified cultural elements and the design of a tourist map, in order to transpose the cultural characteristics of historical cities (in this tourist map). With the example of the historical city of Foshan, China the authors succeed in presenting regional cultural elements scientifically and systematically in a well-designed tourist map.

The paper addresses an interesting topic for the scientific community in the fields of culture, tourism but also for the general public.

My only concerns refer to the fact that the degree of understanding of the created map, the impact it could have, the usefulness for tourists etc. it is not scientifically proven.  However, I think that the paper is well written and documented.

I am suggesting a few points the authors might like to consider in order to improve the manuscript. 

Comments and Suggestions for Authors

Introduction

- Affected by the COVID-19 pandemic, people's physical and mental health were harmed to varying degrees, while outdoor tourism has a positive effect on their recovery [5,6]. I think outdoor tourism refers more to hiking, visiting a national park, general outdoor sports/activities etc. and less activity involved by cultural tourism. Maybe the general term of tourism might be used.

Materials and Methods: 2.1. Research Area

there are 17 national 4A or 5A level tourist attractions and 14 national intangible cul-tural heritage. Could you present what is the difference between the levels and maybe to present some of the attractions as examples?  Is it possible to highlight them in figure 1 (b)?

2.3.2. Determination of Map Extent of TPOI

- Figure 3 - the selected graph type is not correct. For example, according to the chart, there are approximately 140 attractions at exactly 20 km from the Ancient temple and about 240 at 60 km etc. I think that the attractions should be represented by points and possibly draw a correlation curve subsequently.

  Date of this review: 12 Oct. 2022

Author Response

Q1. Introduction- Affected by the COVID-19 pandemic, people's physical and mental health were harmed to varying degrees, while outdoor tourism has a positive effect on their recovery [5,6]. I think outdoor tourism refers more to hiking, visiting a national park, general outdoor sports/activities etc. and less activity involved by cultural tourism. Maybe the general term of tourism might be used.

R1. We totally agree with your suggestion and correct it in the corresponding sentence.

Q2. Materials and Methods: 2.1. Research Area there are 17 national 4A or 5A level tourist attractions and 14 national intangible cultural heritage. Could you present what is the difference between the levels and maybe to present some of the attractions as examples?  Is it possible to highlight them in figure 1 (b)?

R2. The quality level of tourist attractions in China is divided into five levels, which are AAAAA, AAAA, AAA, AA and A tourist attractions from high to low. The difference between 5A and 4A is that 4A can only represent national standard scenic spots, while 5A represent world-class quality. The 5A scenic spot evaluation is more strict and detailed, for example, there are rigid regulations on transportation, tour guides, health, reception and other aspects. In general, scenic spots that can reach these two levels are generally considered to have good service quality and are worth visiting. For instance, Foshan Ancestral Temple in the center of the front of the map is a national 4A scenic spot.

Since Figure 1 (b) has more than 200 points and the figure is so small, the effect of highlighting the location of these scenic spots is not ideal.

Q3. Determination of Map Extent of TPOI- Figure 3 - the selected graph type is not correct. For example, according to the chart, there are approximately 140 attractions at exactly 20 km from the Ancient temple and about 240 at 60 km etc. I think that the attractions should be represented by points and possibly draw a correlation curve subsequently.

R3. Thank you for correcting our mistakes. In fact, the number of attractions can only be an integer, so it is more practical to make a point chart. We have replaced the graphic on page 6 as you suggested.